civil engineering/materials science/inorganic chemistry

mud-cake, solidification agent, metakaolin, geopolymer, bonding strength, oil well

**Authors for correspondence:**
Jiapei Du
e-mail: s3757163@student.rmit.edu.au
Huajie Liu
e-mail: liuhuajieupc@163.com

# Utilization of metakaolin-based geopolymer as a mud-cake solidification agent to enhance the bonding strength of oil well cement–formation interface

Yuhuan Bu[1,2], Rui Ma[2], Jiapei Du[3], Shenglai Guo[1,2], Huajie Liu[1,2] and Letian Zhao[4]

[1]Key Laboratory of Unconventional Oil & Gas Development (China University of Petroleum (East China)), Ministry of Education, Qingdao 266580, People's Republic of China
[2]School of Petroleum Engineering, China University of Petroleum (East China), Qingdao 266580, People's Republic of China
[3]Civil and Infrastructure Discipline, School of Engineering, Royal Melbourne Institute of Technology, Victoria 3001, Australia
[4]Himile Mechanical Science and Technology Corporation (Shandong), Weifang 261500, People's Republic of China

JD, 0000-0002-4309-9528; SG, 0000-0002-5881-4942

This research work designed a novel mud-cake solidification method to improve the zonal isolation of oil and gas wells. The calculation methodology of mud-cake compressive strength was proposed. The optimal formula of activator and solid precursors, the proper activating time and the best activator concentration were determined by the compressive strength test. The effects of solid precursors on the properties of drilling fluid were evaluated. Test results show that the respective percentage of bentonite, metakaolin, slag and activator is $1:1:0.3:0.8$, as well as the optimum ratio of $Na_2SiO_3/NaOH$ is $40:1$. The optimum concentration of activator is 0.21 and the activating time should be more than 10 min. The solid precursors did not show any bad influence on the rheological property of drilling fluids. Even though the compressive strength decreased when the solid precursors blended with barite, the strength values can still achieve 8 MPa. The reaction of metakaolin and activator formed cross-link structure in the mud-cake matrix, which enhanced the connection of the loose bentonite particles, lead to the significant enhancement of shear bonding strength and hydraulic bonding strength. This mud-cake solidification method provides a new approach to improve the quality of zonal isolation.

# 1. Introduction

Cementing operation is an essential part of well construction in the petroleum industry. The quality of zonal isolation is critical for well integrity, and the oil and gas leakage along the well should be avoided [1]. Achieving effective zonal isolation for the life cycle of oil wells depends on the sealing capability of the formation–cement interface. Mud-cake is a thin and impermeable cake formed by the filtration of drilling fluid during the drilling operation. Even though mud-cake can protect the formation from the invasion of drilling fluid during the drilling operation, it becomes an undesirable drawback to offer good zonal isolation [2,3]. In the presence of mud-cake, the cement cannot form an effective bond with the formation [4]. Plenty of studies have been performed to reveal the detrimental effect of mud-cake. Since 1995, Griffith & Osisanya [5] optimized the thickness of drilling fluid filter cakes to enhance the filtrate control capacity of cement slurry and achieve long-term zonal isolation. Bailey *et al*. [6] presented the laboratory test data on the strength of mud-cake and its relationship to wellbore integrity and reservoir damage. They indicate that the pressure differential, mud solids and mud type may be the factors to influence the filter cake yield strength. Ladva *et al*. [7] indicated the key factors, which are pressure, temperature, the permeability of formation, the flexibility of cement, that determine the failure and bonding strength of formation–cement interface. They also proposed some solutions for effective zonal isolation. Opedal *et al*. [4] quantified the effects of rock formation types and drilling fluid formulae on the bonding strength of formation–cement interface. Meanwhile, lots of researchers have investigated the removal of the mud-cake. Zain & Sharma [8] proposed several methods to clean up the wall-building mud-cake. They suggested that the cake removal is not only related to permeability but also depends on mineralogy. Rostami & Nasr-El-Din [9] introduced a novel cleaning approach for mud-cake using a self-destructing water-based fluid. This fluid is weighted with calcium carbonate and both functions of completion and drilling fluid. It has the ability to effectively stimulate the whole well sections after drilling. Al-Arfaj & Amanullah [10] described a guiding tool for the evaluation and testing of different cleaning fluids to select the best one and exclude the inferior. Even though many efforts have been made on the removal of mud-cake, it is still difficult to be cleaned up completely by chemical washes and spacers [11]. Therefore, it is imperative to design a method that can form a mud-cake that can provide an effective bonding between the formation and the cement sheath.

In 1992, Cowan *et al*. [12] proposed the mud-cake to cement (MTC) technology. They converted a water-based drilling fluid to a cement using a hydraulic blast furnace slag. This method improved zonal isolation. However, Benge & Webster [13] raised concerns that MTC cemented body displayed serious brittleness and showed significant risks of collapse during fracture operation. Gu *et al*. proposed the mud-cake to agglomerated cake (MTA) technology in 2009 [14] and then optimized this method in 2011 [15] and 2013 [16]. This method enhanced the zonal isolation of shallow wells. In 2015 [17] and 2016 [2], Gu *et al*. overcame the drawback of MTA technology. They used mud-cake solidification agents (MCSA) as a prepad fluid to make the mud-cake solidified and to enhance the shear bonding strength and cohesive force of mud-cake. Nevertheless, as the main component of mud-cake is bentonite and the reactivity of bentonite is low, the cohesive force of MCSA solidified mud-cake is limited. Thus, this study proposed a novel method to improve the cohesive force of solidified mud-cake significantly. The drilling fluid was prepared with solid precursors to form latent activity mud-cake. The latent activity mud-cake is a kind of mud-cake that will solidify when it encounters an activating agent. The activating agent was used as the prepad fluid to wash the wellbore and to improve the bonding strength of cement formation.

The solid precursors should keep stability before activation. But when these materials connect with a specific activating approach, the reaction process should be fast due to the contacting time of prepad fluid and mud-cake is very short [2]. Geopolymer is a kind of alkali-activated materials [18], which obtains from the alkaline activation of metakaolin [19], fly ash [20] or blast furnace slag [21]. Before activation, these materials cannot form an effective cementitious structure. But when exposed to an alkaline solution, they will form N-A-S-(H) ($Na_2O-Al_2O_3-SiO_2-H_2O$) gel or C-A-S-H ($CaO-Al_2O_3-SiO_2-H_2O$) gel [22] in a short time, which shows high chemical durability and fast strength development. In general, a metakaolin-based geopolymer tends to be more stable because it does not generate calcium hydroxide [23]. Therefore, metakaolin was used in this study as solid precursors, which blended with bentonite to prepare drilling fluid. After the drilling operation, the alkaline solution was pumped into the downhole and then the cement slurry was injected into the annulus.

This research work designed a novel mud-cake solidification method to improve zonal isolation of oil and gas wells. The optimal formula of activator and solid precursor, the proper activating time and the

**Table 1.** Chemical composition of metakaolin and slag.

| | component (wt%) | | | | | | | | |
|---|---|---|---|---|---|---|---|---|---|
| | CaO | SiO$_2$ | FeO | Al$_2$O$_3$ | SO$_3$ | MgO | Na$_2$O | K$_2$O | loss on ignition |
| metakaolin | 0.17 | 55.06 | 0.76 | 42.12 | 0.15 | 0.06 | 0.06 | 0.55 | 1.20 |
| slag | 36.57 | 28.30 | 0.83 | 13.16 | 1.65 | 7.58 | 0.49 | 0.50 | 9.65 |

**Table 2.** Chemical nature, function, dosage and their definition of drilling fluid additives.

| additive | chemical nature | function | dosage (%) |
|---|---|---|---|
| barite | barium sulfate | adjust the density of drilling fluid | 50 |
| SMP | sulfonated-pheno-formoldehyde resin | fluid loss additive | 0–5 |
| SPN4 | walchowite | fluid loss additive | 0–5 |
| CMC | carboxymethylcellulose | fluid loss additive | 1 |
| PAC-AV | vinyl polycopolymers | thickening agent | 0–2 |
| XC | xanthan gum | thickening agent | 0–0.4 |
| XY-28 | zwitterionic polymer | viscosity breaking agent | 0–1 |
| DYFT | sulfonated bitumen | shale-control agent | 0–3.5 |
| AS | aluminium stearate | defoamer | 0–2 |

best concentration of activator were selected by compressive strength test. Rheological property tests were performed to evaluate the effects of solid precursors on the properties of drilling fluid. The performances of this novel method were measured by calculating compressive strength, shear bonding strength and hydraulic bonding strength tests. Scanning electron microscopy (SEM) analysis was used to illustrate the differences of microstructure between solidified mud-cake and unsolidified mud-cake. This method can enhance the shear bonding strength and hydraulic bonding strength of the cement–formation interface significantly. And it almost showed no bad influence on the rheological property of drilling fluid.

## 2. Material and methods

### 2.1. Materials

In this study, the metakaolin was obtained from Jiaozuo Yukun Mining Corporation, China. Slag was provided by Jinan Steel Corporation, China. The chemical composition of metakaolin and slag, which is provided by the supplier and determined by X-ray fluorescence analysis, is shown in table 1. Sodium silicate and sodium hydroxide were supplied by Sinopharm Chemical Reagent Co., Ltd. The additives of drilling fluid were obtained from Shengli Oil Field. Their dosage in drilling fluid, chemical nature and function are shown in table 2.

### 2.2. Optimization of solid precursors and activator

The metakaolin-based solid precursors used in this investigation were made by blending metakaolin and slag. The activator was manufactured by mixing sodium silicate and sodium hydroxide. As the bentonite, which is the main constituent of drilling fluid, possesses low reactivity, the solid precursors should solidify the bentonite effectively when contacting with activators. Therefore, the compressive strength of the solidified bentonite was evaluated. Note that it is better to use cube samples to evaluate the strength property than mud-cake samples. Therefore, in this section, the cube moulds were used and no drilling fluid mud-cake formed. The preparation of the solidified bentonite samples is as follows.

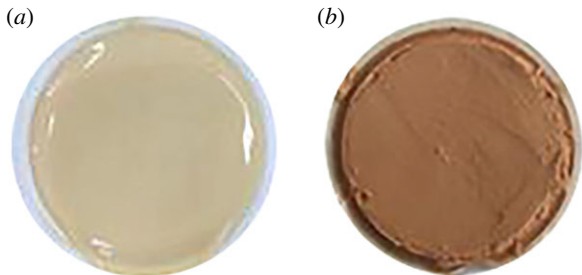

**Figure 1.** Mud-cake samples obtained from base drilling fluid (*a*) and modified drilling fluid (*b*).

Firstly, the weight of metakaolin, slag, bentonite, activator and water was measured in different vessels. Then, water was added in a blending cup at a stirring rate of 2000 r.p.m. When the blender was running, the activator was gradually added into water and mixing for 60 s. After that, metakaolin, slag and bentonite were added into the solution, and another 60 s at a rate of 4000 r.p.m. should be applied to achieve a homogeneous mixture. When the preparation was finished, the mixtures were cast into 5 cm cube moulds and cured at 75°C for 24 h. Compressive strength tests were performed on the samples with different dosages of solid precursors and activators.

## 2.3. Preparation of solidified mud-cake

After the optimization of solid precursors and activator, the solidified mud-cake can be made. The mud-cake samples were prepared under 0.7 MPa at room temperature for 24 h using base drilling fluid (figure 1*a*) and solid precursors modified drilling fluid (figure 1*b*), respectively. The solid precursors modified drilling fluid means than a certain dosage of solid precursor is added into the base drilling fluid to make the drilling fluid able to be activated. As the drilling fluid has been modified by metakaolin and slag, the slag and metakaolin were already in the mud-cake. The mud-cake can be activated using activator. The activator is a pure liquid material, so it is easy to invade the mud-cake. To simulate the real mud-cake activating process in the wellbore, the percolation method was used to activate the mud-cake. The aqueous solution of sodium silicate and sodium hydroxide, with a weight ratio of 40 : 1, was used as an activator. The dosage of activator is varied from 5 to 40%. The activator solution was forced under the pressure of 1 MPa through the mud-cake at 75°C for 6 min. After activation, the modified mud-cake became solidified, but the normal mud-cake unchanged. Then, the solidified mud-cakes were used to perform the mechanical performance test.

## 2.4. Conversion of compressive strength

As the thickness of mud-cake is very thin, the compressive strength of mud-cake cannot be measured directly. A method to convert the break stress value (force of acupuncture by a needle) to the compressive strength of mud-cake (figure 2) was used. Firstly, the acupuncture forces of when mud-cake broke were measured by a specific device (figure 2 (1)). The device was modified by a Vicat apparatus. The force of acupuncture by the needle on the Vicat apparatus can be measured after modification. The conversion from acupuncture force to compressive strength has been well established in the previous study [24]. Subsequently, the finite model of mud-cake was built using the parameters of the real mud-cake. Then, as the mud-cake is considered as a kind of elastic material, the Mises criterion was used to determine the break of mud-cake. According to calculating the Mises stress on a particular path of the mud-cake model, the maximum value of Mises stress on this path was defined as the compressive strength of mud-cake. In this process, the acupuncture force, thickness, area, Young's moduli and Poisson's ratio of real mud-cake are required. Thus, Young's modulus and Poisson's ratio were calculated by the dynamic method [25]. After that, a series of compressive strength values were calculated and the relationship between acupuncture forces and calculated compressive strength can be described by equation (2.1).

$$S = 0.6249F - 0.0079, \tag{2.1}$$

where S is the calculated compressive strength of mud-cake, MPa; F is the acupuncture force of when mud-cake broke, N.

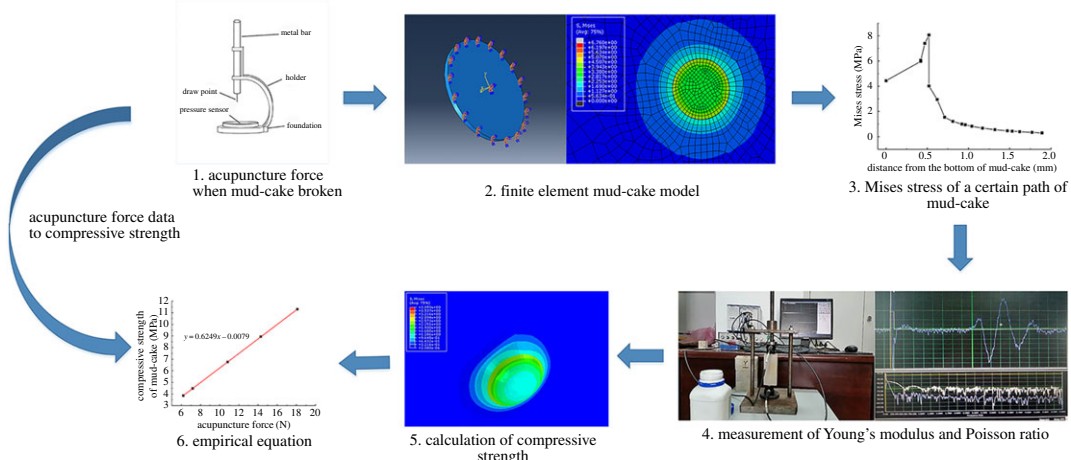

**Figure 2.** Conversion method of mud-cake compressive strength.

**Table 3.** Mix design parameters and designation of drilling fluid.

| drilling fluid | formulation |
| --- | --- |
| TEST-1 | 4% base mud + 5% potassium chloride + 2% DYFT + 1% CMC + 0.1% XC |
| TEST-2 | 4% base mud + 1% PAC-AV + 1.5%SMP + 1%CMC + +0.1% XC + 1% SPN4 + 50% barite |
| H4-1 | weighted drilling fluid |
| SLKL | low water loss drilling fluid |

## 2.5. Shear bonding strength test

In the oil and gas industry, the shear bonding strength of the cement–formation interface is an important parameter to evaluate the sealing capability of the cement–formation interface. To evaluate the influence of mud-cake solidification agent on the cement–formation interface, the shear bonding strength and hydraulic bonding strength tests were performed. Four kinds of drilling fluids were used in this test. The mix design parameters and their designation are shown in table 3. These formulae are provided to evaluate the effects of different kinds of drilling fluid on the mechanical properties of solidified mud-cake. As the oil field required us to keep the formula H4-1 and SLKL confidential, the mix design was not provided. The samples used for shear bonding strength were prepared as follows. Firstly, the core was put into the filtration apparatus to make mud-cake covered on its surface. Then, distilled water was used to clean up the false filter cake and drilling fluid on the surface of the core. Subsequently, the activator solution was forced under the pressure of 1 MPa through the core at 75°C. After that, the mud-cake covered core was placed into a steel cylinder and cement slurry was pulled into the annulus between the core and the steel cylinder. The samples were cured at 75°C for 24 h. The schematic of the shear bonding strength test is shown in figure 3. An incremental load was used on the press cake to move the core out of the cement sheath. Then a peak value of the load obtained corresponded to the breaking of the interface between the core and cement. As the mud-cake is very thin, the thickness of the mud-cake is negligible. Thus, the shear bonding strength can be deduced as equation (2.2).

$$P = \frac{5F}{\pi RH},$$ (2.2)

where P is the shear bonding strength, MPa; F is the maximum value of the load, kN; R is the radius of the core and H is the height of the core, cm.

## 2.6. Hydraulic bonding strength test

The hydraulic bonding strength is an essential parameter to evaluate the quality of the cement–formation interface. It represents the ability of the cement–formation interface to resist water penetration [7].

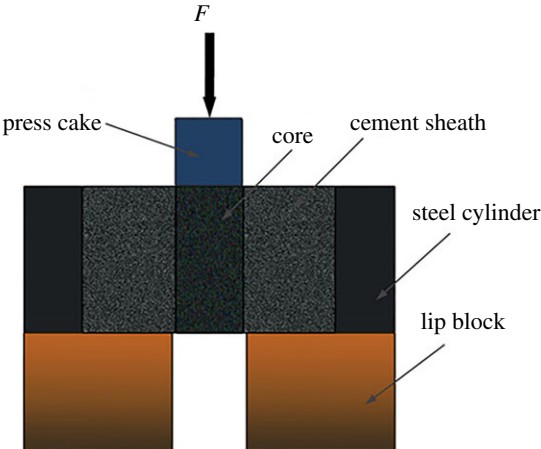

**Figure 3.** Schematic of shear bonding strength test.

The samples used for hydraulic bonding strength tests were prepared as follows. Firstly, sand was filled in the filtration apparatus at a certain height and then water was used to saturate the sand. After that, the activated mud-cake was placed into the sand. Subsequently, cement slurry was pulled into the space between mud-cake and the wall of the filtration apparatus. Note that the height of cement slurry should be lower than the height of the mud-cake to make sure that the water can reach the mud-cake. After 24 h curing at 75°C, hydraulic pressure was added from the top of the apparatus. The water flowed past the cement–formation interface and flowed out from the bottom of the kettle. The pressure value when the bottom water flow increasing suddenly was recorded. Therefore, this pressure value can be considered as the hydraulic bonding strength of the cement–formation interface. The mechanism of hydraulic bonding strength test is shown in figure 4.

## 2.7. SEM analyses

The typical mud-cake sample and solidified mud-cake sample, which was cured at 75°C for 24 h, were selected to perform SEM analysis. Specific areas were picked and focused with different magnifications at a voltage of 20 kV to distinguish possible differences between normal mud-cake and solidified mud-cake.

# 3. Results and discussion

## 3.1. Compressive strength of blended samples

### 3.1.1. Effect of metakaolin dosage

The compressive strength test results of blended samples with different dosages of metakaolin are shown in figure 5. The percentages of slag and activator were 10% and 60%, respectively, by weight of bentonite. When the dosage of metakaolin is lower than 100% by weight of bentonite, the compressive strength increases significantly with the increase of metakaolin dosage. While the dosage of metakaolin is more than 100%, the compressive strength increases slightly. Due to a large amount of metakaolin that might have a terrible influence on the properties of drilling fluid, the dosage of 100% was selected as the optimum dosage of metakaolin.

### 3.1.2. Effect of slag dosage

Figure 6 shows the compressive strength test results of blended samples with different dosages of slag. The percentages of metakaolin and activator were 100% and 60%, respectively, by weight of bentonite. The compressive strength values of blended samples increase with the rise of slag dosage when the dosage of slag was less than 30%. Subsequently, the compressive strength values showed a noticeable decrease while the dosage of slag was more than 30%. The partial substitute of metakaolin with slag increased the compressive strength attribute to the formation of C-A-S-H after the addition of slag with high calcium content. The C-A-S-H gel fills the voids in N-A-S-H gel, reducing the

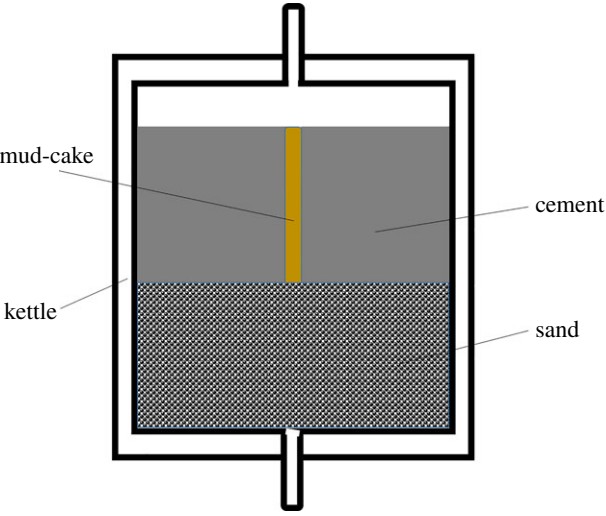

**Figure 4.** Mechanism of hydraulic bonding strength test.

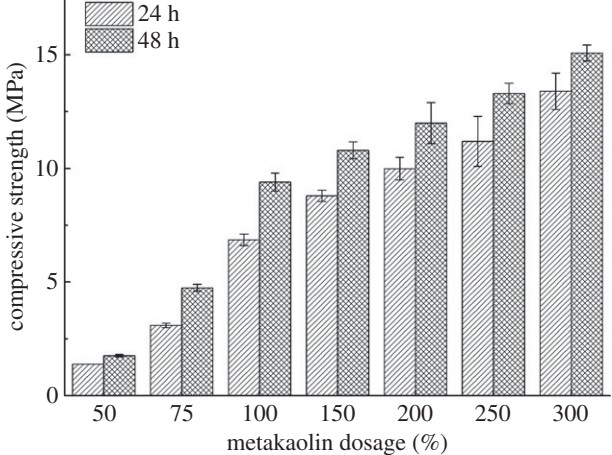

**Figure 5.** Compressive strength tests results of blended samples with different dosage of metakaolin.

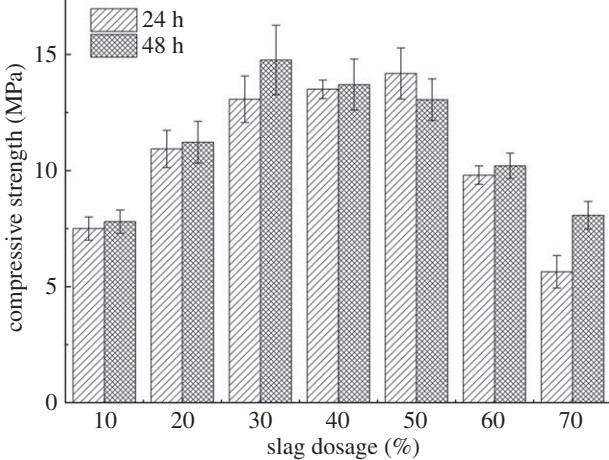

**Figure 6.** Compressive strength tests results of blended samples with different dosage of slag.

permeability and porosity, resulting in an increase of compressive strength [18,26–28]. However, the excessive slag formed superfluous calcium hydroxide, which led to a rapid decrease in compressive strength [29]. The maximum strength value of slag-enhanced samples reached 15 MPa for 48 h, which is

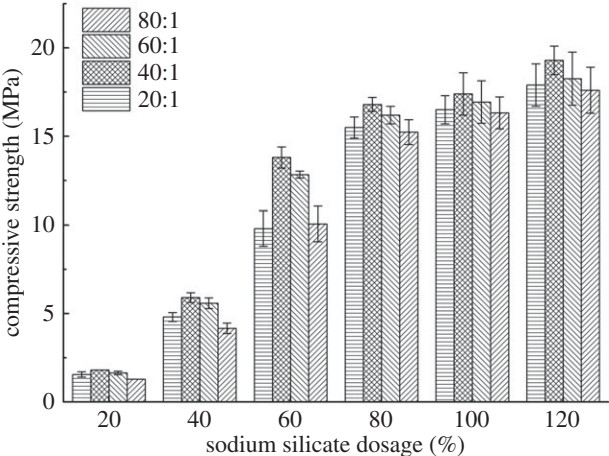

**Figure 7.** Compressive strength tests results of blended samples with different dosage of sodium silicate (by weight of binder) and Na$_2$SiO$_3$/NaOH ratio.

much larger than the samples without slag (9.4 MPa). Thus, the optimum dosage of slag is 30% by weight of bentonite.

### 3.1.3. Effect of activator dosage

The effect of sodium silicate dosage and Na$_2$SiO$_3$/NaOH ratio on the compressive strength of blended samples is shown in figure 7. The percentages of metakaolin and slag are 100% and 30%, respectively, by weight of bentonite. The variation of compressive strength is very sensitive to the dosage of sodium silicate when the dosage is changing from 20% to 80%. When the dosage of sodium silicate is more than 80%, the strength enhances slightly with the increase of sodium silicate dosage. Meanwhile, even though the Na$_2$SiO$_3$/NaOH ratio was changed, the variation of compressive strength still showed the same trend. This phenomenon illustrates that, when sodium silicate is used to activate the solid precursors, the dosage of sodium silicate should be more than 80% to make the mud-cake achieve enough strength. For the consideration of economic factors, the optimum dosage of sodium silicate is 80% by weight of bentonite.

As for the weight ratio of Na$_2$SiO$_3$/NaOH, with the increasing of Na$_2$SiO$_3$/NaOH ratio, the compressive strength of blended samples increases first and then decreases. The compressive strength value reaches the maximum when the Na$_2$SiO$_3$/NaOH ratio is 40:1, regardless of the variation of sodium silicate dosage. Therefore, after the step-by-step optimization, the respective percentage of bentonite, metakaolin, slag and activator is 1:1:0.3:0.8. And the optimum ratio of Na$_2$SiO$_3$/NaOH is 40:1.

## 3.2. Calculated compressive strength of solidified mud-cake

### 3.2.1. Effect of activator concentration

As the percolation method was used to activate the mud-cake, the concentration of activator in solution is a critical factor to the final solidified strength of mud-cake. Thus, the effect of activator concentration in solution on calculated compressive strength of solidified mud-cake was studied. The concentration of activator, which is defined as the weight ratio of activator and water, was varied from 0.05 to 0.41. See from figure 8, the calculated compressive strength of solidified mud-cake increases sharply with the rise of activator concentration when the activator concentration is lower than 0.21. After that, the increasing trend of compressive strength becomes slow when the concentration of activator is more than 0.21, due to the limited proportion of solid precursors in mud-cake. Hence, the optimum concentration of the activator is 0.21.

### 3.2.2. Effect of activating time

The calculated compressive strength of solidified mud-cake after different activating time is shown in figure 9. With the increase of activating time, the calculated compressive strength of mud-cake shows

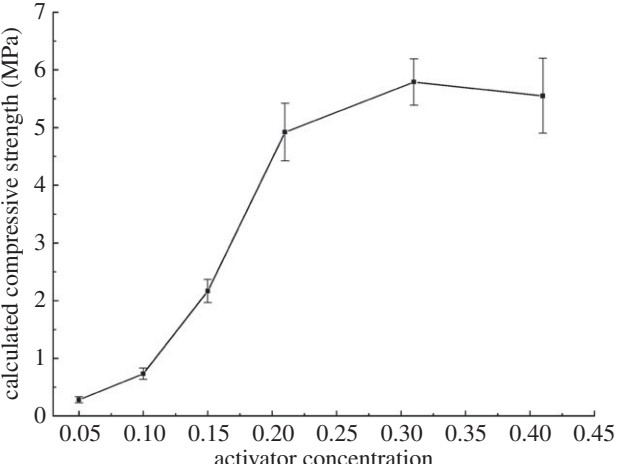

**Figure 8.** Calculated compressive strength of solidified mud-cake under different activator concentrations.

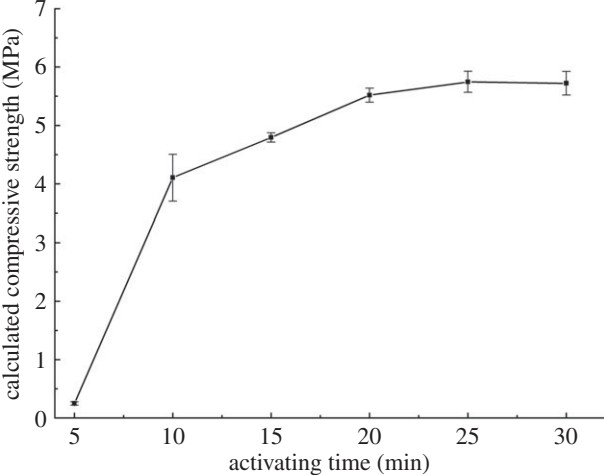

**Figure 9.** Calculated compressive strength of solidified mud-cake after different activating time.

a significant increase when the activating time is less than 10 min. After the activating time reaches 10 min, the increasing trend of compressive strength becomes slower. This phenomenon illustrates that the activating time should be more than 10 min.

## 3.3. Effect of solid precursors on rheological property of drilling fluid

### 3.3.1. Effects of different kinds of drilling fluid

The rheological property is vital for drilling fluid to carry debris, deliver pump power and ensure the drilling speed [30]. After the doping of solid precursors, the solid phase of drilling fluid increased significantly. Thus, it is necessary to evaluate the influence of solid precursors on the rheological property of drilling fluids. The effects of solid precursors on the rheological property of two kinds of drilling fluid are shown in figure 10. As we can see, the pure SLKL and H4-1 drilling fluid show the characteristics of Bingham fluid. After the drilling fluid modified by solid precursors, the modified SLKL and H4-1 drilling fluid still show characteristics of Bingham fluid. The shearing force of modified drilling fluid increases slightly due to the increase of the solid phase. Hence, the solid precursors did not show any bad influence on the rheological property of drilling fluids.

### 3.3.2. Effects of pH value

In general, downhole conditions affect the rheological property of drilling fluid. The pH value of downhole varies in different well sections. Sodium hydroxide was used to adjust the pH value of

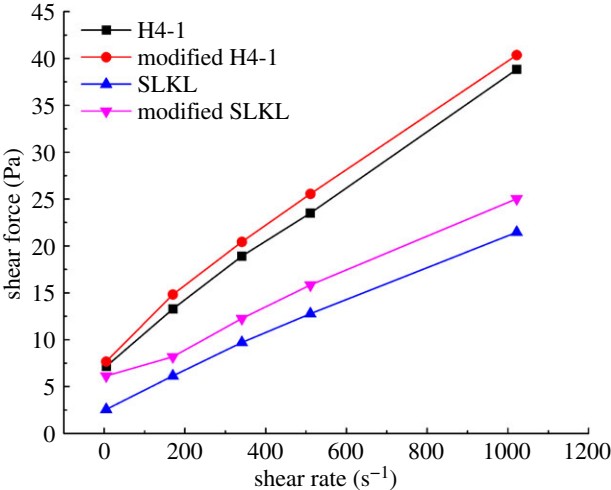

**Figure 10.** Effects of solid precursors on rheological property of different kinds of drilling fluid.

modified drilling fluid. And the prepared drilling fluid was digested for 16 h at 80°C. The effects of pH value on the rheological property of drilling fluid are shown in table 4. As we can see, when the pH value is changing from 9 to 11, the shearing force does not show any difference and the shearing rate shows a slight change. Thus, the pH value of drilling fluid shows no significant effect on the modified drilling fluid.

### 3.3.3. Effects of circulation time

The rheological property of drilling fluid after a long-time shearing at the downhole temperature should be considered. The modified drilling fluid shearing for 0, 16 and 48 h at 80°C were studied, as shown in table 5. The shearing rate and shearing force both show a slight increase with the rise of circulation. After long-time circulation, the part of the solid precursors was activated and the consistency of modified drilling fluid increased. However, the rheological property of modified drilling fluid still meets the requirement of field operation.

### 3.3.4. Effects of temperature

The temperature in the downhole condition influences the activation of solid precursors. The activation of solid precursors leads to an increase of the consistency of drilling fluid. The modified drilling fluids were digested at 20, 80, 120 and 140°C, respectively, for 16 h. Table 6 displays the effects of circulation temperature on the rheological property of drilling fluid. When the circulation temperature changed from 20 to 120°C, the shearing force of modified drilling fluid was unchanged and the shearing rate showed a slightly increase. At 140°C, the shearing force and shearing rate showed significant increase. This is due to the activity of solid precursors being activated by high temperatures, resulting in the formation of large hydration particles. Therefore, the suitable operation temperature of the modified drilling fluid is from 20 to 140°C.

## 3.4. Compatibility with drilling fluid

### 3.4.1. Effects of barite dosage

Barite is an important admixture to adjust the density of drilling fluids and to keep the pressure stability in the wellbore [31,32]. As barites were commonly used in large quantity when adjusting the density of drilling fluid, the effects of barite dosage on compressive strength of solidified mud-cake should be considered. Figure 11 presents the compressive strength variation with the changing of barite dosages. As we can see, when the dosage of barite is less than 20%, the compressive strength of solidified mud-cake decreases significantly. This is attributed to barite not possessing cementitious activity. While the drilling fluid containing barite was used to prepare solidified mud-cake, interfaces were formed between cementitious phases and barite. These interfaces lead to low compressive strength of solidified mud-cake. When the dosage of barite was more than 20%, with the increase of barite

**Table 4.** Effects of pH value on the rheological property of H4-1 drilling fluid.

| pH | $\Phi600$ | $\Phi300$ | $\Phi200$ | $\Phi100$ | $\Phi6$ | $\Phi3$ | PV | $\tau_0$ |
|---|---|---|---|---|---|---|---|---|
| 9.2 | 8 | 5 | 4 | 3 | 1 | 1 | 3 | 1.022 |
| 10 | 8 | 5 | 4 | 4 | 2 | 1 | 3 | 1.022 |
| 11 | 10 | 6 | 3.5 | 2 | 1 | 1 | 4 | 1.022 |

**Table 5.** Effects of circulation time on rheological property of H4-1 drilling fluid.

| circulation time (h) | $\Phi600$ | $\Phi300$ | $\Phi200$ | $\Phi100$ | $\Phi6$ | $\Phi3$ | PV | $\tau_0$ |
|---|---|---|---|---|---|---|---|---|
| 0 | 8 | 5 | 4 | 3 | 1 | 1 | 3 | 1.022 |
| 16 h | 12 | 7 | 6 | 3 | 1 | 1 | 5 | 1.022 |
| 48 h | 15 | 9 | 7 | 5 | 4 | 4 | 6 | 1.533 |

**Table 6.** Effects of circulation temperature on rheological property of H4-1 drilling fluid.

| circulation temperature (°C) | $\Phi600$ | $\Phi300$ | $\Phi200$ | $\Phi100$ | $\Phi6$ | $\Phi3$ | PV | $\tau_0$ |
|---|---|---|---|---|---|---|---|---|
| 20 | 8 | 5 | 4 | 3 | 1 | 1 | 3 | 1.022 |
| 80 | 12 | 7 | 6 | 3 | 1 | 1 | 5 | 1.022 |
| 120 | 14 | 8 | 5 | 3 | 1 | 1 | 6 | 1.022 |
| 140 | 24 | 14 | 10 | 5 | 1 | 1 | 10 | 2.044 |

dosage, the compressive strength of solidified mud-cake decreases. Even though the compressive strength decreased, the strength values still achieved 8 MPa.

### 3.4.2. Effects of additives

Drilling fluid additives, including fluid loss additive, thickening agent, viscosity breaking agent, shale-control agent and defoamer, are used to adjust the properties of drilling fluid [33]. Some kinds of additives can influence the solidification of mud-cake. The effects of additives on the compressive strength of solidified mud-cake are shown in figure 12. The thickening agent of XC, XY-28 and PAC-AV, the fluid loss additives of SPN4, SMP and the shale-control agent of DYFT were selected to perform the investigation. Figure 12 illustrates that the thickening agent of XC and XY-28 showed a slight effect on the compressive strength of mud-cake, due to the dosage of these two additives being very low in drilling fluid. On the contrary, the thickening agent of PAC-AV, the fluid loss additives of SMP, SPN4 and the shale-control agent of DYFT showed a more serious influence on the compressive strength of solidified mud-cake. But the compressive strength values of these samples are still larger than 10 MPa, which indicates that if the dosages of these additives are controlled in the commonly used interval, the compressive strength of mud-cake can reach more than 10 MPa. Therefore, the influence of drilling fluid additives is not significant on the compressive strength of solidified mud-cake.

## 3.5. Shear bonding strength of cement–formation interface

Normally, the property of the cement–formation interface can be evaluated by the shear bonding strength of the cement–formation interface. Figure 13 shows the shear bonding strength of the cement–formation interface, which was made by different drilling fluids. As we can see, the shear bonding strength showed significant enhancement after modification by solid precursors. The same variation can be observed in different samples, which were made by different kinds of drilling fluid. The shear bonding strength of

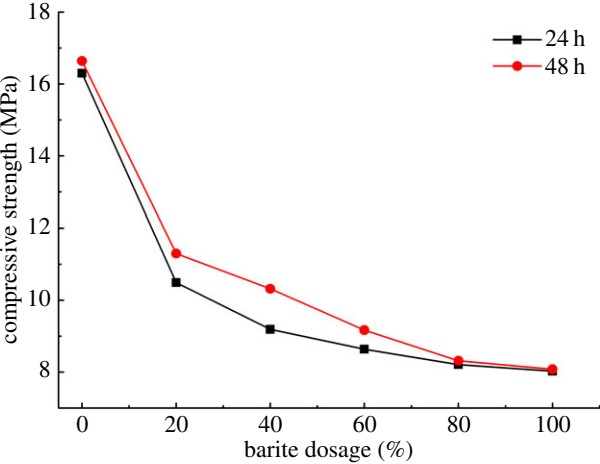

**Figure 11.** Effects of barite dosage on compressive strength of mud-cake.

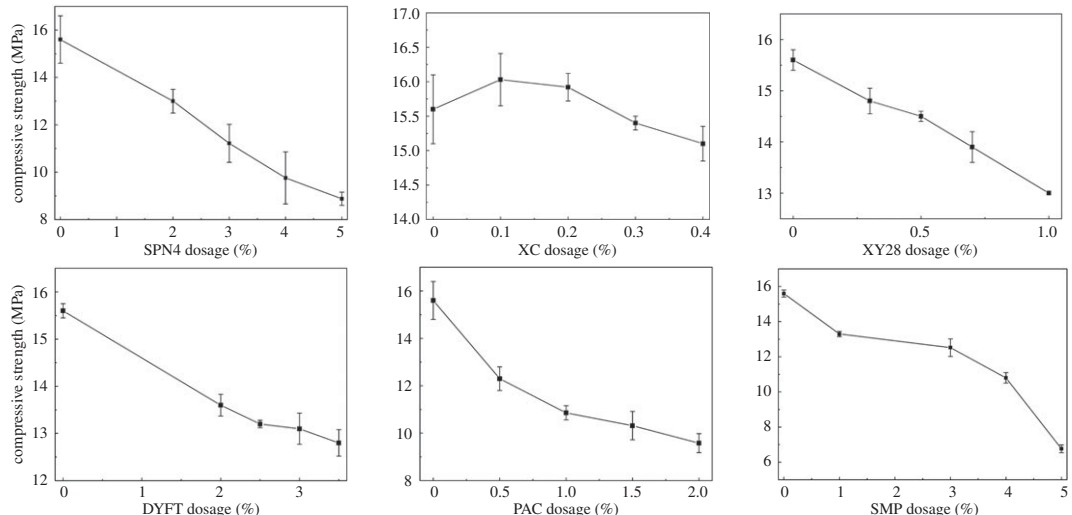

**Figure 12.** Effects of drilling fluid additives on the compressive strength of solidified mud-cake.

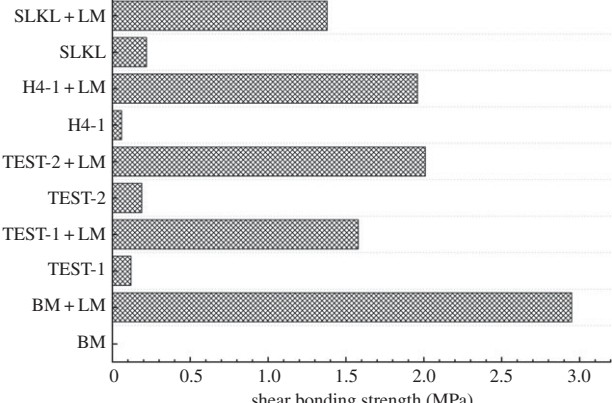

**Figure 13.** Shear bonding strength of cement–formation interface.

modified samples was improved by 7 times, 33 times, 11 times and 13 times for modified SLKL, H4-1, TEST-2 and TEST-1 samples, respectively, compared to unmodified samples. Meanwhile, the strength values are also much higher than the values in references [34,35]. Thus, this method can improve the bonding strength of cement–formation interface significantly.

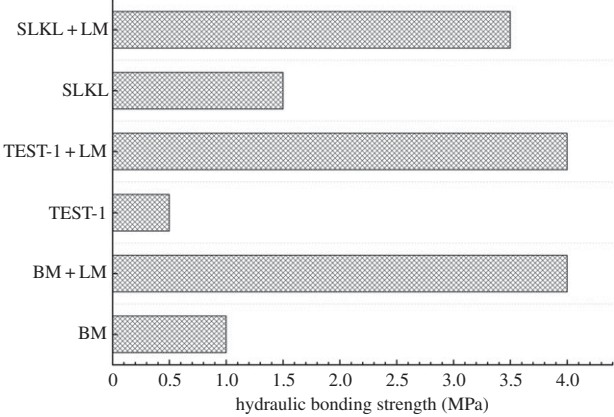

**Figure 14.** Hydraulic bonding strength of cement–formation interface.

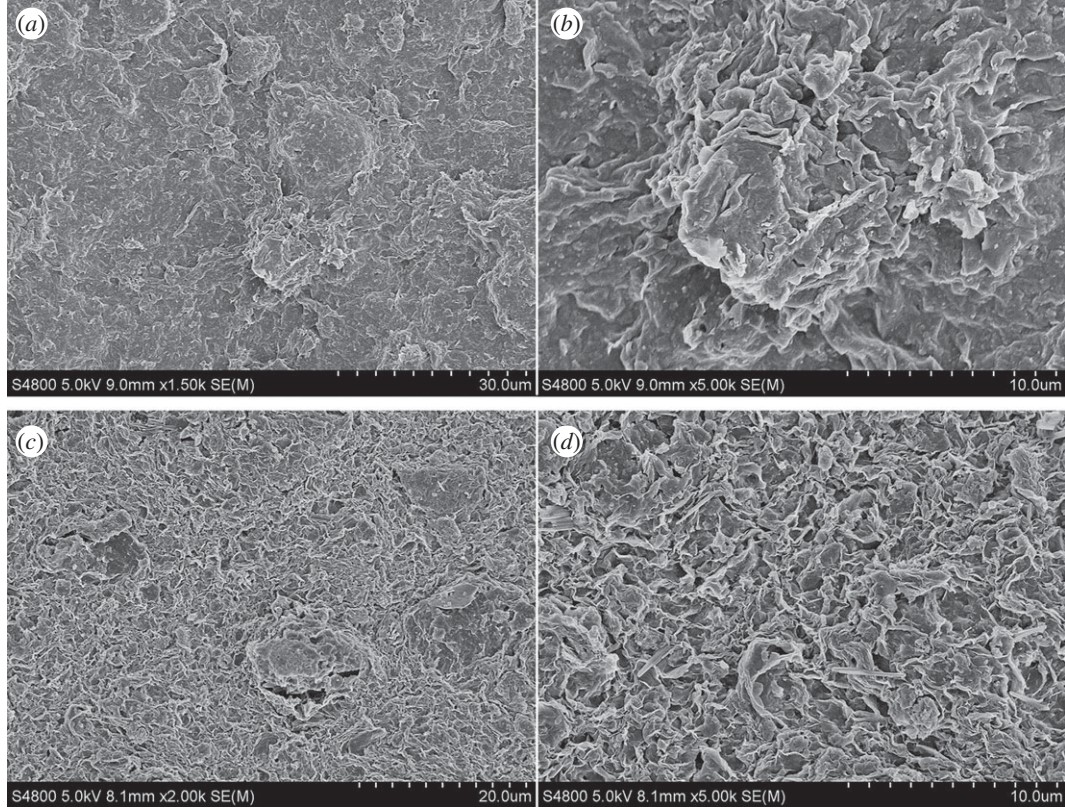

**Figure 15.** SEM images of (*a*) and (*b*) unmodified mud-cake sample; (*c*) and (*d*) modified mud-cake sample.

## 3.6. Hydraulic bonding strength

The hydraulic bonding strength is another vital parameter to evaluate the bonding strength of the cement–formation interface. This value shows the ability of the cement–formation interface to resist water and gas channelling. The method of hydraulic bonding strength tests is illustrated in §2.6. As the permeability of cement stone is very low and the bonding strength of the cement–kettle interface can reach 7 MPa, which is much more than the hydraulic bonding strength value, this method can properly characterize the isolation capacity of fluid at cement–formation interface. As shown in figure 14, the hydraulic bonding strength of solidified samples improved greatly, compared to the unsolidified samples. The strength values of solidified samples were enhanced by two times, eight times and four times for modified SLKL, TEST-1 and BM, respectively. Hence, the mud-cake solidification method can improve the isolation capacity of fluid at the cement–formation interface effectively.

### 3.7. SEM analysis

The micromorphology of the surface of the unsolidified mud-cake sample and solidified mud-cake sample is shown in figure 15. As we can see, although the unsolidified mud-cake shows a compact structure when the magnification is 1500×, the structure becomes loose when the magnification changes to 5000×. On the contrary, the solidified mud-cake displays disordered structure while the magnification is 2000×, but while the magnification becomes 5000×, the mud-cake matrix shows a cross-linking structure. This different morphology is due to the reaction of metakaolin and activator that formed the geopolymer gel. The geopolymer gel enhanced the connection of the loose bentonite particles and improved the strength of mud-cake.

## 4. Conclusion

In this study, a novel mud-cake solidification method to improve zonal isolation of oil and gas wells was designed. Metakaolin-based geopolymer was used as a mud-cake solidification agent. Based on the results of the test, the following conclusions can be derived:

(1) The results of compressive strength of blended samples show that the respective percentage of bentonite, metakaolin, slag and activator is $1:1:0.3:0.8$, as well as the optimum ratio of $Na_2SiO_3/NaOH$ is $40:1$. The partial substitute of metakaolin with slag increased the compressive strength is attributed to the formation of C-A-S-H in slag with high calcium content. But the excess slag formed a superfluous dosage of calcium hydroxide, which led to a rapid decrease in compressive strength.

(2) Analysis of calculated compressive strength display that the optimum concentration of activator is 0.21 due to the limited proportion of solid precursors in mud-cake, and the activating time should be more than 10 min.

(3) The solid precursors did not show any obvious bad influence on the rheological property of drilling fluids in different conditions. Even though the compressive strength decreased when the solid precursors blended with barite or other drilling fluid additives, the strength values still achieved 8 MPa.

(4) The reaction of metakaolin and activator formed cross-link structure in the mud-cake matrix, which enhanced the connection of the loose bentonite particles, led to the significant enhancement of shear bonding strength and hydraulic bonding strength.

Data accessibility. Data available from the Dryad Digital Repository: https://doi.org/10.5061/dryad.9rh2873 [36].
Authors' contribution. Y.B. carried out the statistical analyses, designed the study, coordinated the study and helped draft the manuscript; J.D. carried out the laboratory work, participated in data analysis, participated in the design of the study and drafted the manuscript; S.G. participated in data analysis work; H.L. helped draft the manuscript; L.Z. carried out the laboratory work. All authors gave final approval for publication.
Competing interests. We declare we have no competing interests.
Funding. This work is supported by National Natural Science Foundation of China (51974355, 51704321) and Program for Changjiang Scholars and Innovative Research Team in University (IRT_14R58).
Acknowledgements. We thank editors and anonymous reviewers for their helpful suggestions on the early versions of this manuscript.

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
