## [Reviewer comments · Royal Society Open Science]

Review History

RSOS-191230.R0 (Original submission)

Review form: Reviewer 1

Is the manuscript scientifically sound in its present form?

Yes

Are the interpretations and conclusions justified by the results?

Yes

Is the language acceptable?

No

Do you have any ethical concerns with this paper?

No

Have you any concerns about statistical analyses in this paper?

No

Recommendation?

Accept with minor revision (please list in comments)

Comments to the Author(s)

This manuscript proposes a utilization of metakaolin based geopolymer as a mud-cake solidification agent to enhance the bonding strength of oil well cement-formation. The subject is interesting to RSOS readers. Here are my comments on this manuscript:

The abstract is too long and it needs to be made shorter and concise.

English writing needs to be improved. There are some grammar and punctuation issues that need to be fixed. Help from a native English speaker is needed.

In page 5 table 1, Fe(II) instead of Fe(III) is probably the chemical state of iron in slag. Please specify the method used to determine the chemical composition of both metakaolin and slag.

In page 5 table 2, it listed a lot of additives with code names only. It is unacceptable for a research article. Please specify the chemical nature of the additives as well as the dosage used in this study.

In page 6-7 section 2.4, what does 'acupuncture force' mean, or it means penetration force? Which kind of device has been used to determine the force? Was the conversion from 'acupuncture force' to 'compressive strength' well established? Please cite related literatures or give more detail explanation on this conversion.

In: "The metakaolin-based solid precursors used in this investigation were made by blending metakaolin and slag. The activator was manufactured by mixing sodium silicate and sodium hydroxide." Why choose metakaolin and slag?

In: "The preparation of the solidified bentonite samples is as follows: Firstly, the weight of metakaolin, slag bentonite, activator and water was measured in different vessels. Then, water was added..." How is the drilling fluid mud-cake formed?

In: "solid precursors modified drilling fluid..." What is solid precursors modified drilling fluid in respect to? Please specify it.

In: "After the optimization of solid precursors and activator, the solidified mud-cake can be made..." As the author mentioned in the introduction: "mud-cake is a thin and impermeable cake formed by the filtration of drilling fluid during...". Thus, how do slag and other materials invade the mud-cake?

Review form: Reviewer 2

Is the manuscript scientifically sound in its present form?

Yes

Are the interpretations and conclusions justified by the results?

Yes

Is the language acceptable?

Yes

Do you have any ethical concerns with this paper?

No

Have you any concerns about statistical analyses in this paper?

No

Recommendation?

Accept with minor revision (please list in comments)

Comments to the Author(s)

Suggestion:

- 1) Line 35: Bailey et al presented the laboratory test data.....Please elaborate more about the laboratory test data.
- 2) Line 36 : Ladva et al indicate the key factors that.....What are the key factors?
- 3) Line 42 : . Rostami et al. introduced a novel approach for.....Explain/describe the novel approach
- 4) Line 93: table 1, Please write all result in 2 decimal point
- 5) Line 195 : Please rearrange this paragraph before fig.5
- 6) Line 205 : Please rearrange this paragraph before fig.6
- 7) Line 222 : Please rearrange this paragraph before fig.7
- 8) Line 243 : Please rearrange this paragraph before fig.8
- 9) Line 275 : Please rearrange this paragraph before Table 4
- 10) Line 284 : Please rearrange this paragraph before Table 5
- 11) Line 292 : Please rearrange this paragraph before Table 6
- 12) Line 305 : Please rearrange this paragraph before fig. 11
- 13) fig 9 &10 :missing in manuscript
- 14) Line 318 : Please rearrange this paragraph before fig 12
- 15) Line 336 : Please rearrange this paragraph before fig 13
- 16) Line 348 : Please rearrange this paragraph before fig 14
- 17) Fig 15 : please provide space/gap between image (a) and (b)
- 18) Line 362 : Please rearrange this paragraph before fig 15
- 19) Line 374 : Please change the numbering from 1 to 4.1, 4.2,4.3 and 4.4

Decision letter (RSOS-191230.R0)

17-Jan-2020

Dear Dr Du

On behalf of the Editors, I am pleased to inform you that your Manuscript RSOS-191230 entitled "Utilization of metakaolin based geopolymer as a mud-cake solidification agent to enhance the bonding strength of oil well cement-formation interface" has been accepted for publication in Royal Society Open Science subject to minor revision in accordance with the referee suggestions. Please find the referees' comments at the end of this email.

The reviewers and handling editors have recommended publication, but also suggest some minor revisions to your manuscript. Therefore, I invite you to respond to the comments and revise your manuscript.

- Ethics statement

- Data accessibility

It is a condition of publication that all supporting data are made available either as supplementary information or preferably in a suitable permanent repository. The data accessibility section should state where the article's supporting data can be accessed. This section should also include details, where possible of where to access other relevant research materials

such as statistical tools, protocols, software etc can be accessed. If the data has been deposited in an external repository this section should list the database, accession number and link to the DOI for all data from the article that has been made publicly available. Data sets that have been deposited in an external repository and have a DOI should also be appropriately cited in the manuscript and included in the reference list.

If you wish to submit your supporting data or code to Dryad (<http://datadryad.org/>), or modify your current submission to dryad, please use the following link:
<http://datadryad.org/submit?journalID=RSOS&manu=RSOS-191230>

- **Competing interests**

- **Authors' contributions**

- **Acknowledgements**

- **Funding statement**

Because the schedule for publication is very tight, it is a condition of publication that you submit the revised version of your manuscript before 26-Jan-2020. Please note that the revision deadline will expire at 00.00am on this date. If you do not think you will be able to meet this date please let me know immediately.

When submitting your revised manuscript, you will be able to respond to the comments made by

the referees and upload a file "Response to Referees" in "Section 6 - File Upload". You can use this to document any changes you make to the original manuscript. In order to expedite the processing of the revised manuscript, please be as specific as possible in your response to the referees. We strongly recommend uploading two versions of your revised manuscript:

If your manuscript is newly submitted and subsequently accepted for publication, you will be asked to pay the article processing charge, unless you request a waiver and this is approved by Royal Society Publishing. You can find out more about the charges at <https://royalsocietypublishing.org/rsos/charges>. Should you have any queries, please contact openscience@royalsociety.org.

on behalf of Professor Ian Guymer (Associate Editor) and R. Kerry Rowe (Subject Editor)
 openscience@royalsociety.org

Associate Editor Comments to Author (Professor Ian Guymer):

Associate Editor: 1

Comments to the Author:

Thank you for your submission to Royal Society Open Science. We have now received two reports for your manuscript; both of which are supportive of publication but request some revisions. Please address these in your point-by-point response letter. Please also provide a marked-up version of your manuscript, which highlights the changes you have made in your revision.

Please also ensure to conduct a full English language check with your revision. As you have been requested to edit the written English, you must provide proof that you have done so: acceptable proof includes a certificate of language-editing from a language editing service or a signed letter from a native speaker of English. If you do not provide this proof, your manuscript will be returned to you.

For information about language editing services endorsed by the Royal Society, please follow the link below:

<https://royalsociety.org/journals/authors/language-polishing/>

Reviewer comments to Author:

Reviewer: 1

Comments to the Author(s)

This manuscript proposes a utilization of metakaolin based geopolymer as a mud-cake solidification agent to enhance the bonding strength of oil well cement-formation. The subject is interesting to RSOS readers. Here are my comments on this manuscript:

The abstract is too long and it needs to be made shorter and concise.

English writing needs to be improved. There are some grammar and punctuation issues that need to be fixed. Help from a native English speaker is needed.

In page 5 table 1, Fe(II) instead of Fe(III) is probably the chemical state of iron in slag. Please specify the method used to determine the chemical composition of both metakaolin and slag.

In page 5 table 2, it listed a lot of additives with code names only. It is unacceptable for a research article. Please specify the chemical nature of the additives as well as the dosage used in this study.

In page 6-7 section 2.4, what does 'acupuncture force' mean, or it means penetration force? Which kind of device has been used to determine the force? Was the conversion from 'acupuncture force' to 'compressive strength' well established? Please cite related literatures or give more detail explanation on this conversion.

In: "The metakaolin-based solid precursors used in this investigation were made by blending metakaolin and slag. The activator was manufactured by mixing sodium silicate and sodium hydroxide." Why choose metakaolin and slag?

In: "The preparation of the solidified bentonite samples is as follows: Firstly, the weight of metakaolin, slag bentonite, activator and water was measured in different vessels. Then, water was added..." How is the drilling fluid mud-cake formed?

In: "solid precursors modified drilling fluid..." What is solid precursors modified drilling fluid in respect to? Please specify it.

In: "After the optimization of solid precursors and activator, the solidified mud-cake can be made..." As the author mentioned in the introduction: "mud-cake is a thin and impermeable

cake formed by the filtration of drilling fluid during...". Thus, how do slag and other materials invade the mud-cake?

Reviewer: 2

Comments to the Author(s)

Suggestion:

- 1) Line 35: Bailey et al presented the laboratory test data.....Please elaborate more about the laboratory test data.
- 2) Line 36 : Ladva et al indicate the key factors that.....What are the key factors?
- 3) Line 42 : . Rostami et al. introduced a novel approach for.....Explain/describe the novel approach
- 4) Line 93: table 1, Please write all result in 2 decimal point
- 5) Line 195 : Please rearrange this paragraph before fig.5
- 6) Line 205 : Please rearrange this paragraph before fig.6
- 7) Line 222 : Please rearrange this paragraph before fig.7
- 8) Line 243 : Please rearrange this paragraph before fig.8
- 9) Line 275 : Please rearrange this paragraph before Table 4
- 10) Line 284 : Please rearrange this paragraph before Table 5
- 11) Line 292 : Please rearrange this paragraph before Table 6
- 12) Line 305 : Please rearrange this paragraph before fig. 11
- 13) fig 9 &10 :missing in manuscript
- 14) Line 318 : Please rearrange this paragraph before fig 12
- 15) Line 336 : Please rearrange this paragraph before fig 13
- 16) Line 348 : Please rearrange this paragraph before fig 14
- 17) Fig 15 : please provide space/gap between image (a) and (b)
- 18) Line 362 : Please rearrange this paragraph before fig 15
- 19) Line 374 : Please change the numbering from 1 to 4.1, 4.2,4.3 and 4.

Author's Response to Decision Letter for (RSOS-191230.R0)

See Appendix A.

Decision letter (RSOS-191230.R1)

31-Jan-2020

Dear Dr Du,

It is a pleasure to accept your manuscript entitled "Utilization of metakaolin based geopolymer as a mud-cake solidification agent to enhance the bonding strength of oil well cement-formation interface" in its current form for publication in Royal Society Open Science. The comments of the reviewer(s) who reviewed your manuscript are included at the foot of this letter.

Please ensure that you send to the editorial office an editable version of your accepted manuscript, and individual files for each figure and table included in your manuscript. You can send these in a zip folder if more convenient. Failure to provide these files may delay the

processing of your proof. You may disregard this request if you have already provided these files to the editorial office.

on behalf of Professor Ian Guymmer (Associate Editor) and R. Kerry Rowe (Subject Editor)
openscience@royalsociety.org

Appendix A

Reviewer comments to Author:

Reviewer: 1

Comments to the Author(s)

This manuscript proposes a utilization of metakaolin based geopolymer as a mud-cake solidification agent to enhance the bonding strength of oil well cement-formation. The subject is interesting to RSOS readers. Here are my comments on this manuscript:

Q1: The abstract is too long and it needs to be made shorter and concise.

Response to Reviewer comment No. 1: Thanks for the reviewer's suggestion.

The abstract has been revised to make it shorter and concise. The previous word count was 217, and now become 194.

“This research work designed a novel mud-cake solidification method to improve the zonal isolation of oil and gas wells. The calculation methodology of mud-cake compressive strength was proposed. The optimal formula of activator and solid precursors, the proper activating time and the best activator concentration were determined by the compressive strength test. The effects of solid precursors on the properties of drilling fluid were evaluated. Test results show that the respective percentage of bentonite, metakaolin, slag and activator is 1:1:0.3:0.8, as well as the optimum ratio of $\text{Na}_2\text{SiO}_3/\text{NaOH}$ is 40:1. The optimum concentration of activator is 0.21 and the activating time should be more than 10 min. The solid precursors did not show any bad influence on the rheological property of drilling fluids. Even though the compressive strength decreased when the solid precursors blended with barite, the strength values can still achieve 8 MPa. The reaction of metakaolin and activator formed crosslink structure in the mud-cake matrix, which enhanced the connection of the loose bentonite particles, lead to the significant enhancement of shear bonding strength and hydraulic bonding strength. This mud-cake solidification method provides a new approach to improve the quality of zonal isolation.”

Q2: English writing needs to be improved. There are some grammar and punctuation issues that need to be fixed. Help from a native English speaker is needed.

Response to Reviewer comment No. 2: Thanks for the reviewer's suggestion.

The English writing has been improved by a professional translation agency. The grammar and

punctuation issues are fixed. The correction proof is attached at the end of this file.

Q3: In page 5 table 1, Fe(II) instead of Fe(III) is probably the chemical state of iron in slag. Please specify the method used to determine the chemical composition of both metakaolin and slag.

Response to Reviewer comment No. 3: Thanks for the reviewer's suggestion.

The Fe₂O₃ in **Table 1** is changed to FeO. The chemical composition information of metakaolin and slag are both provided by the supplier. The chemical composition was determined by X-ray fluorescence analysis. This is clarified in section **2.1 Materials**.

“2.1 Materials

*In this study, the metakaolin was obtained from Jiaozuo Yukun Mining Corporation, China. Slag was provided by Jinan Steel Corporation, China. The chemical composition of metakaolin and slag, which is provided by the supplier and determined by X-ray fluorescence analysis, is shown in **Table 1**. ”*

Table 1 Chemical composition of metakaolin and slag

	Component (Wt.%)								
	CaO	SiO ₂	FeO	Al ₂ O ₃	SO ₃	MgO	Na ₂ O	K ₂ O	Loss on ignition
Metakaolin	0.17	55.06	0.76	42.12	0.15	0.06	0.06	0.55	1.2
Slag	36.57	28.3	0.83	13.16	1.65	7.58	0.49	0.5	9.65

Q4: In page 5 table 2, it listed a lot of additives with code names only. It is unacceptable for a research article. Please specify the chemical nature of the additives as well as the dosage used in this study.

Response to Reviewer comment No. 4: Thanks for the reviewer's suggestion.

The chemical nature and dosage of the additives are specified in Table 2.

Table 2 Chemical nature, function, dosage and their definition of drilling fluid additives

Additive	Chemical nature	Function	Dosage
barite	Barium sulfate	Adjust the density of drilling fluid	50%
SMP	Sulfonated-pheno-formoldehyde	Fluid loss additive	0-5%

	resin		
SPN4	Walchowite	Fluid loss additive	0-5%
CMC	Carboxymethylcellulose	Fluid loss additive	1%
PAC- AV	Vinyl polycopolymers	Thickening agent	0-2%
XC	Xanthan gum	Thickening agent	0-0.4%
XY-28	Zwitterionic polymer	Viscosity breaking agent	0-1%
DYFT	Sulfonated bitumen	Shale-control agent	0-3.5%
AS	Aluminum stearate	defoamer	0-2%

Q5: In page 6-7 section 2.4, what does 'acupuncture force' mean, or it means penetration force? Which kind of device has been used to determine the force? Was the conversion from 'acupuncture force' to 'compressive strength' well established? Please cite related literatures or give more detail explanation on this conversion.

Response to Reviewer comment No. 5: Thanks for the reviewer's suggestion.

The acupuncture force means that the stress value of acupuncture by a needle when the mud-cake break. It is measured by a specific device, which modified by a Vicat apparatus, as shown in Fig.2(1). The conversion from acupuncture force to compressive strength has already been well established. This is clarified in section 2.4 and related literature is cited.

“2.4 Conversion of compressive strength

Due to the thickness of mud-cake is very thin, the compressive strength of mud-cake cannot be measured directly. A method to convert the break stress value (force of acupuncture by a needle) to the compressive strength of mud-cake (Fig.2) was used. Firstly, the acupuncture forces of when mud-cake broke were measured by a specific device (Fig.2 (1)). The device was modified by a Vicat apparatus. The force of acupuncture by the needle on the Vicat apparatus can be measured after modification. The conversion from acupuncture force to compressive strength has been well established in previous study [24].”

Reference

[24] Bu Y, Zhou L, Du J, Guo B, Zhao L. A compressive strength evaluation method of solidified mud cake in cement-formation interface. *Research and Exploration in Laboratory*. 2019, 38(1):

Fig.2(1)

Q6: In: "The metakaolin-based solid precursors used in this investigation were made by blending metakaolin and slag. The activator was manufactured by mixing sodium silicate and sodium hydroxide." Why choose metakaolin and slag?

Response to Reviewer comment No. 6: Thanks for the reviewer's suggestion.

The metakaolin and slag are alkaline activation materials. Before activation, these materials cannot form an effective cementitious structure. But when exposed to an alkaline solution, they will form effective cementitious structure in a short time. This characteristic perfectly satisfies the demand of mud-cake solidification. This illustration also displayed in the **Introduction** section.

"Geopolymer is a kind of alkali-activated materials [19], which obtains from the alkaline activation of metakaolin [20], fly ash [21] or blast furnace slag [22]. Before activation, these materials cannot form an effective cementitious structure. But when exposed to an alkaline solution, they will form N-A-S-(H) ($\text{Na}_2\text{O}-\text{Al}_2\text{O}_3-\text{SiO}_2-\text{H}_2\text{O}$) gel or C-A-S-H ($\text{CaO}-\text{Al}_2\text{O}_3-\text{SiO}_2-\text{H}_2\text{O}$) gel [23] in a short time, which shows high chemical durability and fast strength development."

Reference:

[19] Hao H, Gu J, Huang J, et al. 2016 Comparative study on cementation of cement-mudcake interface with and without mud-cake-solidification-agents application in oil & gas wells. *J. Petrol. Sci. Eng.* 147, 143-53. (doi: 10.1016/j.petrol.2016.05.014)

[20] Alcamand HA, Borges PHR, Silva FA, et al. 2018 The effect of matrix composition and calcium content on the sulfate durability of metakaolin and metakaolin/slag alkali-activated mortars. *Ceram Int.* 44(5), 5037-5044. (doi: 10.1016/j.ceramint.2017.12.102)

[21] Barbosa VFF, MacKenzie KJD, Thaumaturgo C. 2000 Synthesis and characterisation of materials based on inorganic polymers of alumina and silica: sodium polysialate polymers. *Int. J. Inorganic Mater.* 2(4), 309-317. (doi: 10.1016/S1466-6049(00)00041-6)

[22] Fernández-Jiménez A, Palomo A. 2005 Composition and microstructure of alkali activated fly ash binder: Effect of the activator. *Cem. Concr. Res.* 35(10), 1984-1992. (doi: 10.1016/j.cemconres.2005.03.003)

[23] Puertas F, Martínez-Ramírez S, Alonso S, et al. 2000 Alkali-activated fly ash/slag cements: Strength behaviour and hydration products. *Cem. Concr. Res.* 30(10), 1625-32. (doi: 10.1016/S0008-8846(00)00298-2)

Q7: In: "The preparation of the solidified bentonite samples is as follows: Firstly, the weight of metakaolin, slag bentonite, activator and water was measured in different vessels. Then, water was added..." How is the drilling fluid mud-cake formed?

Response to Reviewer comment No. 7: Thanks for the reviewer's suggestion.

This section aims to select the activator of mud-cake. Due to bentonite is the main constituent of drilling fluid mud-cake, the chemical reaction among bentonite, metakaolin, slag and activator was evaluated. It is better to use cube samples to evaluate the strength property than mud-cake samples. Therefore, in this section, the cube molds were used and no drilling fluid mud-cake formed. And this is also clarified in section **2.2 Optimization of solid precursors and activator.**

"Due to the bentonite, which is the main constituent of drilling fluid, possesses low reactivity, the solid precursors should solidify the bentonite effectively when coming across activators. Therefore, the compressive strength of solidified bentonite was evaluated. Note that it is better to use cube samples to evaluate the strength property than mud-cake samples. Therefore, in this section, the cube molds were used and no drilling fluid mud-cake formed."

Q8: In: "solid precursors modified drilling fluid..." What is solid precursors modified drilling fluid in respect to? Please specify it.

Response to Reviewer comment No. 8: Thanks for the reviewer's suggestion.

The solid precursors modified drilling fluid means than a certain dosage of solid precursor is added into the base drilling fluid to make the drilling fluid can be activated. This is clarified in section **2.3 Preparation of solidified mud-cake.**

"2.3 Preparation of solidified mud-cake

After the optimization of solid precursors and activator, the solidified mud-cake can be made. The mud-cake samples were prepared under 0.7 MPa at room temperature for 24 h using base drilling fluid and solid precursors modified drilling fluid. The solid precursors modified drilling fluid means than a certain dosage of solid precursor is added into the base drilling fluid to make the drilling fluid can be activated.

Q9: In: "After the optimization of solid precursors and activator, the solidified mud-cake can be made..."As the author mentioned in the introduction: "mud-cake is a thin and impermeable cake formed by the filtration of drilling fluid during...". Thus, how do slag and other materials invade the mud-cake?

Response to Reviewer comment No. 9: Thanks for the reviewer's suggestion.

Due to the drilling fluid has already been modified by metakaolin and slag, the slag and metakaolin were already in the mud-cake. The mud-cake can be activated by using activator. The activator is a pure liquid material, so it is easy to invade the mud-cake. This is specified in section **2.3 Preparation of solidified mud-cake.**

"2.3 Preparation of solidified mud-cake

After the optimization of solid precursors and activator, the solidified mud-cake can be made. The mud-cake samples were prepared under 0.7 MPa at room temperature for 24 h using base drilling fluid (Fig.1 left) and solid precursors modified drilling fluid (Fig.1 right). The solid precursors modified drilling fluid means than a certain dosage of solid precursor is added into the base drilling fluid to make the drilling fluid can be activated. Due to the drilling fluid has been modified by metakaolin and slag, the slag and metakaolin were already in the mud-cake. The mud-cake can be activated by using activator. The activator is a pure liquid material, so it is easy to invade the mud-cake.

Reviewer: 2

Comments to the Author(s)

Suggestion:

Q1: Line 35: Bailey et al presented the laboratory test data.....Please elaborate more about the laboratory test data.

Q2: Line 36 : Ladva et al indicate the key factors that.....What are the key factors?

Q3: Line 42 : . Rostami et al. introduced a novel approach for.....Explain/describe the novel approach.

Response to Reviewer comment No. 1-3: Thanks for the reviewer's suggestion.

More detailed information about the literature are given in the Introduction section.

“Bailey et al. presented the laboratory test data on the strength of mud-cake and its relationship to wellbore integrity and reservoir damage [6]. They indicate that the pressure differential, mud solids and mud type may be the factors to influence the filtercake yield strength. Ladva et al. indicated the key factors, which are pressure, temperature, the permeability of formation, the flexibility of cement, that determine the failure and bonding strength of formation-cement interface. They also proposed some solutions for effective zonal isolation [7]. Opedal et al. quantified the effects of rock formation types and drilling fluid formulas on the bonding strength of formation-cement. Meanwhile, lots of researchers have investigated the removal of the mud-cake [4]. Zain and Sharma proposed several methods to clean up the wall-building mud cake. They suggested that the cake removal is not only related to permeability but also depends on mineralogy [8]. Rostami et al. introduced a novel approach for mud-cake cleaning utilizing a self-destructing water-based fluid [9]. This fluid is weighted with calcium carbonate and both functions of completion and drilling fluid. It has the ability to effectively stimulate the whole well sections after drilling.”

Q4: Line 93: table 1, Please write all result in 2 decimal point

Response to Reviewer comment No. 4: Thanks for the reviewer's suggestion.

All results in Table 1 are corrected in 2 decimal point.

Table 1 Chemical composition of metakaolin and slag

	Component (Wt.%)								
	CaO	SiO ₂	FeO	Al ₂ O ₃	SO ₃	MgO	Na ₂ O	K ₂ O	Loss on ignition
Metakaolin	0.17	55.06	0.76	42.12	0.15	0.06	0.06	0.55	1.20
Slag	36.57	28.30	0.83	13.16	1.65	7.58	0.49	0.50	9.65

Q5: Line 195 : Please rearrange this paragraph before fig.5

Q6: Line 205 : Please rearrange this paragraph before fig.6

Q7: Line 222 : Please rearrange this paragraph before fig.7

Q8: Line 243 : Please rearrange this paragraph before fig.8

Q9: Line 275 : Please rearrange this paragraph before Table 4

Q10: Line 284 : Please rearrange this paragraph before Table 5

Q11: Line 292 : Please rearrange this paragraph before Table 6

Q12: Line 305 : Please rearrange this paragraph before fig. 11

Response to Reviewer comment No. 5-12: Thanks for the reviewer's suggestion.

All of these paragraphs are rearranged.

Q13: fig 9 &10 :missing in manuscript

Response to Reviewer comment No. 13: Thanks for the reviewer's suggestion.

Fig 9 and 10 are added in manuscript.

Q14: Line 318 : Please rearrange this paragraph before fig 12

Q15: Line 336 : Please rearrange this paragraph before fig 13

Q16: Line 348 : Please rearrange this paragraph before fig 14

Response to Reviewer comment No. 14-16: Thanks for the reviewer's suggestion.

All of these paragraphs are rearranged.

Q17: Fig 15 : please provide space/gap between image (a) and (b)

Response to Reviewer comment No. 17: Thanks for the reviewer's suggestion.

The gap has been provided in Fig.15.

Q18: Line 362 : Please rearrange this paragraph before fig 15

Response to Reviewer comment No. 18: Thanks for the reviewer's suggestion.

The paragraph before Fig.15 has been rearranged.

Q19: Line 374 : Please change the numbering from 1 to 4.1, 4.2,4.3 and 4.4

Response to Reviewer comment No. 19: Thanks for the reviewer's suggestion.

The numbering is changed to 4.1, 4.2, 4.3 and 4.4.